# Interaction with a Virtual Coach for Active and Healthy Ageing

**DOI:** 10.3390/s23052748

**Published:** 2023-03-02

**Authors:** Michael McTear, Kristiina Jokinen, Mirza Mohtashim Alam, Qasid Saleem, Giulio Napolitano, Florian Szczepaniak, Mossaab Hariz, Gérard Chollet, Christophe Lohr, Jérôme Boudy, Zohre Azimi, Sonja Dana Roelen, Rainer Wieching

**Affiliations:** 1School of Computing, Ulster University, Belfast BT15 1AP, UK; 2Artificial Intelligence Research Center, National Institute of Advanced Industrial Science and Technology (AIRC/AIST), Tokyo 135-0064, Japan; 3Institut für Angewandte Informatik (INFAI), 04109 Leipzig, Germany; 4Institut Mines-Télécom (IMT), 91120 Palaiseau, France; 5Institut für Experimentelle Psychophysiologie Gmbh (IXP), 40215 Düsseldorf, Germany; 6Chair for Business Information Systems and New Media, Faculty III, Universität Siegen (USI), 57068 Siegen, Germany

**Keywords:** active and healthy ageing, dialogue system, knowledge sources, sensors, participatory design

## Abstract

Since life expectancy has increased significantly over the past century, society is being forced to discover innovative ways to support active aging and elderly care. The e-VITA project, which receives funding from both the European Union and Japan, is built on a cutting edge method of virtual coaching that focuses on the key areas of active and healthy aging. The requirements for the virtual coach were ascertained through a process of participatory design in workshops, focus groups, and living laboratories in Germany, France, Italy, and Japan. Several use cases were then chosen for development utilising the open-source Rasa framework. The system uses common representations such as Knowledge Bases and Knowledge Graphs to enable the integration of context, subject expertise, and multimodal data, and is available in English, German, French, Italian, and Japanese.

## 1. Introduction

With life expectancy increasing dramatically over the last century, societies are becoming increasingly confronted with new difficulties associated with an ageing population and the need to develop smart living solutions for the care of older adults and the promotion of active and healthy ageing [1]. This paper describes an innovative virtual coaching technique that is presently being developed in a three-year joint European (H2020) and Japanese (MIC) research project [2]. The e-VITA project has received funding from the European Union H2020 Programme under grant agreement no. 101016453. The Japanese consortium received funding from the Japanese Ministry of Internal Affairs and Communication (MIC), Grant no. JPJ000595. With the objective of enabling older persons to better manage their own health and daily activities, the virtual coach provides support and motivation in the critical areas of active and healthy ageing in cognition, physical activity, mobility, mood, social interaction, leisure, and spirituality. This will lead to greater well-being and enhanced stakeholder engagement. The virtual coach will provide individualized profiling and personalized recommendations based on big data analytics and social-emotional computing, detecting risks in the user’s daily living environment by collecting data from external sources and non-intrusive sensors, and providing support through natural interactions with 3D-holograms, emotional objects, or robotic technologies using multimodal and spoken dialogue technology, and advanced Knowledge Graph and Knowledge Base research. Ref. [3] presents a summary of the project in general.

The rest of the paper is organized as follows. Section 2 examines relevant research in the field of active and healthy ageing, with a focus on the use of conversational systems to provide a virtual coaching application. Section 3 discusses the technologies used in the e-VITA virtual coach, including sensor-based multimodal data fusion, emotion recognition, knowledge graphs and knowledge bases, and conversational artificial intelligence (AI). Section 4 discusses the creation of an early prototype system, which included a collaborative design approach that resulted in use cases and content development, some of which was included in a first prototype system. Section 5 closes by discussing the future phases of the project.

## 2. Related Work

There has been increasing interest in recent years in exploring how new developments in AI and related technologies can be applied to support active and healthy ageing in older adults.

One important line of research involves the use of visual systems to monitor the movement, behaviour, and health of older adults. Ref. [4] describes a study involving the analysis of videos of elderly people performing three mobility tests and how the videos could be automatically categorized using deep neural networks in order to emulate how they are evaluated by expert physiotherapists. It was found that the best results were achieved by a deep Conv-BiLSTM classifier. Ref. [5] reviews the use of motion capture sensors for musculoskeletal health monitoring and explores methods for transforming motion capture data into kinematics variables and factors that affect the tracking performance of RGB-D sensors, concluding that RGB-D sensor-based approaches offer an inexpensive and viable option for the provision of contactless healthcare access to patients. Ref. [6] explores the issue of monitoring the activities of older adults in multi-resident environments, describing how a sensor network using a Bluetooth Low Energy (BLE) signal emitted by tags that are worn by residents as well as passive infrared (PIR) motion sensors can be used to monitor the location and activities of individual residents.

Video-based approaches were considered in the early stages of the e-VITA project to detect non-verbal emotions through the capture of gestures and facial expressions, but due to internal project time constraints and the requirement to develop a first Virtual Coach prototype early in the project, we focused more on speech-based emotion capture (see Section 3.2) and the capture of actimetric behavioural data with mobile and fixed environmental domotic sensors such as accelerometers, gyroscopes, and Presence Infrared Sensors (PIR) (see Section 3.1). The aim is to integrate this work with the next version of the virtual coach.

In this paper, we focus in particular on how new developments in conversational AI and socially assistive robots can be applied to support active and healthy ageing. Ref. [7] presents a comprehensive state-of-the-art review in which several future research directions are identified, including the need for unconstrained natural language processing and conversational strategies to enable robust and meaningful two-way conversations; the ability to interpret affective modalities in order to enhance user engagement and trust; and challenges in deployment, ensuring user adherence and data privacy. Looking at some relevant individual projects, Ref. [8] reports a month-long study of a virtual agent and robot that could interact with older adults in their homes through dialogue and gestures, with the aim of providing companionship and reducing isolation. The importance of dialogue capability to enable social robot agents to provide natural interaction is also emphasised in Ref. [9], while Ref. [10] describes how information about the older adult’s emotional status was extracted from an analysis of their verbal and non-verbal communication. Ref. [11] is an example of an ongoing project in the Netherlands with the aim of improving the lives of older adults through the use of voice technology. Particularly relevant to the e-VITA virtual coach, Ref. [12] describes the Dialogue Management and Language Generation components of a voice-based spoken dialogue system that can conduct challenging and complex coaching conversations with older adults. The system was successfully tested and validated in user studies with older adults in Spain, France, and Norway, with three different languages and cultures.

There have been several funded research projects in the European Union and in Japan that have been concerned with active and healthy ageing and that are relevant to the e-VITA project.

HOLOBALANCE [13] developed an augmented reality coaching system for balance training using wearable sensors to capture movement while performing exercises;CAPTAIN (Coach Assistant via Projected and Tangible Interface) [14] used a range of different innovative technologies such as projected augmented reality, 3D sensing technologies, tangible user interaction, and physiological and emotional data analysis to turn all surfaces into tangible interfaces for personalized information and reminders. Coaching interventions included reminders about medicine intake or suggesting physical training exercises as well as warnings, for example, that the oven was turned on;WellCO (Wellbeing and Health Virtual Coach) [15] provides personalized advice and guidance to users in relation to the adoption of healthier behaviour choices using data from wearable sensors and visual and speech emotion recognition;SAAM (Supporting Active Ageing through Multimodal Coaching) [16] assists older persons in their everyday tasks and wellbeing at home by collecting health, emotional, and cognitive data from sensors, smart meters, and other devices, providing recommendations for a healthy diet and hobbies, and suggesting social and cultural events based on the preferences and interests of the individual person;vCare [17] recommends coaching activities for the rehabilitation of older people after treatment in hospital based on their personalized care pathways, engaging with them to encourage compliance;NESTORE [18] offers personalized advice and supports decision-making for older adults to promote wellbeing and independent living. There is a chatbot interface that accepts text-based input from users in English, Spanish, Italian, and Dutch, and detects emotion from the user’s text;ACCRA [19] was a joint European–Japanese initiative that includes two social robots. Astro is an assistive robotic platform dedicated to mobility and user interaction using natural language, a touch screen, and a visual LED system, while Buddy is a small robot that is designed to act as a companion at home;CARESSES [20] is jointly funded by the European Commission and the Ministry of Internal Affairs and Communications of Japan. The project uses social robots to assist people with reminders to take medication, encouraging them to keep active, and helping them keep in touch with family and friends. A particular emphasis of the project was to pay attention to the customs, cultural practices, and individual preferences of the older adults;EMPATHIC [21] is concerned with developing a personalized virtual coach to enable older adults to live independently by engaging in extended conversations about potential chronic diseases, maintaining a healthy diet and physical activity, and encouraging social engagement. The technical aspects of the project focus on innovative multimodal face and speech analytics and adaptive spoken dialogue systems involving senior citizens in Spain, Norway, and France.

Some of these projects, such as HOLOBALANCE and CAPTAIN, also address vision-based approaches, for instance in order to recognize human body postures or to perform a gait analysis (e.g., for fall prevention), or alternatively, as in the WellCO, NESTORE and EMPATHIC projects, in order to track facial emotion.

The e-VITA project also focuses on the issues addressed in these examples of related work but in addition aims to extend the research to examine how data from sensors and emotion analysis can be used within dialogues with the older users, and how knowledge graphs and knowledge bases can be used to store data relevant to active healthy ageing as well as personal data about the user to enable the virtual coach to offer personalized information and recommendations.

## 3. Technologies and Architecture of the Virtual Coach

The complete e-VITA platform is based on an extended version of the Digital Enabler platform provided by the project partner Engineering (Italy) [22]. The platform is designed to include multiple devices and software components from across Europe and Japan that are based on different technologies and standards. The platform supports communication and integration among different smart devices such as sensors and robots, as well as the collection and management of data to provide coaching functionalities.

### 3.1. Multimodal Data Fusion from Sensors

In a smart home, sensors can be used to capture and monitor data in order to provide the user with safe and comfortable assisted living. In order to provide this information and in the framework of the e-VITA project, our approach is that of data fusion. Data fusion is both intuitive and complex to define. Here we will adopt the following definition [23]:

Data fusion is a formal framework in which means and tools for the alliance of data originating from different sources are expressed. It aims at obtaining information of greater quality; the exact definition of ‘greater quality’ will depend upon the application.

Our various data sources come from the different sensors present in the environment. These can be of three types:Sensors worn by the user (embedded systems);Those giving information on the environment and which are not worn (thermometer, barometer, etc.);The sensors specific to the smart home (PIR, opening detector).

In addition, we must consider their nature. Indeed, a sensor can be “continuous” (after its activation, the sensor sends information continuously and at a given frequency, e.g., as an accelerometer) or “event-driven” (the sensor sends information only under certain conditions, for example, as a presence detector). We can then establish the following (non-exhaustive) classification, as shown in Table 1.

The formal framework can belong to one of two categories: algorithmic or architectural, as shown in Figure 1.

The algorithmic approach is an efficient and traditional approach because it allows everything to be centralized. Moreover, it provides new information by taking into account all multimodal data at the same time. Nevertheless, all sources must be simultaneously available at any time. In a smart home context, the user can choose whether or not to equip their home with different sensors. It is also possible that the sensors in a smart home may fail and/or that the sensors worn by the person are simply omitted. This conventional approach cannot support a situation where one or more modules are missing [24].

The architectural approach allows us to see each source as an agent and to route these agents to a centralized information node. This node will first store the incoming data to ensure its availability. Then, this information will be sent to the corresponding service for a specific use. This mechanism is described in Figure 1 by the links between the data collector, the central data node, and the data fusion algorithm. In the context of a smart home, this approach compensates for the drawbacks of the algorithmic approach. Thus, we have proposed a novel architectural approach (based on the components of the FIWARE standard) [25], the principle being based on an interweaving and communication of the algorithmic approaches. Our architecture follows the following (simplified) scheme, as shown in Figure 2, where the dotted arrows represent specific data flows from the home and smartphone acquisition stages towards a data storage memory server.

Thanks to this architecture it is possible to collect data from different sources (home and smartphone data collector—based on the Django framework), then centralize them (Orion context broker). Note that the architecture is able to preserve any modification of the data by making a chronicle of them (performed by the Cygnus historisation service depicted in Figure 2, the FIWARE standard). Moreover, by using the subscriptions (the notification mechanism of the broker), we can trigger, on demand, different machine learning algorithms present in the computing unit, e.g., Long Short-Term Memory (LSTM) for the actimetric analysis from the accelerometer and the gyroscope data of the smartphone. The latter can retrieve the data directly via the subscription or via STH-Comet (time series querying). The objective is to provide actimetric data, specific to each user, to a dialogue manager based on Rasa.

Machine learning algorithms were used to determine the user’s actimetrics. The current literature provides a substantial number of articles dealing with the recognition of human activity (HAR) or lack of activity in a smart home environment. Ref. [26] highlighted the capabilities of a coupled Convolutional Neural Network–Long Short-Term Memory network (CNN-LSTM) in the classification of human activity from the signals of the gyroscope and accelerometer of the smartphone. Moreover, if our environment is that of a smart home or a living lab, Ref. [27] shows us three approaches for classifying human activity in this context. These two articles have in common that they use the public datasets from the University of California Irvine (https://archive.ics.uci.edu/ml/datasets.php) (accessed on 5 January 2023). At this point, it is possible to take advantage of this investigation in order to integrate them in our previous fusion architecture. Thus, our system is able to provide labels from the different user’s situations or states to the dialogue manager, enabling it to adapt to the different contexts at its disposal.

### 3.2. Emotion Detection System

Affective or emotional computing is the creation of emotionally-aware technology that analyses affective and expressive behaviour [28]. Emotions can be detected in various acoustic parameters in the users’ voice, including the frequency, time, amplitude, and spectral energy domains [29]. The Emotion Detection System (EDS) in the e-VITA platform primarily uses features from the spectral energy domain to detect and classify emotions during interactions between the coaching system and older adults [30]. The detected emotions can then be used by the dialogue system to provide appropriate interventions [28].

The e-VITA emotion detection system (EDS) is a deep neural network model for speech emotion classification, constructed from a two-dimensional convolutional neural network (2D-CNN) and a long short-term memory (LSTM) network. The model is designed to use a log-mel spectrogram to learn both local and global features. The log-mel spectrogram provides a compact representation of the frequency content of an audio signal and is robust to noise and recording variations. To prepare the input for the LSTM layer, the log-mel spectrogram is split into smaller frames of equal size (window size of 128 and a hop size of 64). The frames allow the LSTM to process the input data sequentially and in a time-dependent manner, which is important for learning patterns in the audio signal. The frame size and step are chosen to capture the relevant temporal information on the audio signal while limiting the number of parameters in the LSTM model.

The model includes four local feature learning blocks (LFLBs) and one LSTM layer. Each LFLB consists of a 2D-CNN to extract temporal and spectral features from the input spectrogram, a batch normalization layer to normalize the output of the CNN layer, max pooling to down-sample the 2D-CNN output and reduce parameters of the model, and a dropout layer to reduce overfitting. The output is fed to the LSTM layer through a flattening layer. The LSTM layer models the temporal dependencies between the local features, and the output layer maps the output of the LSTM layer to a probability distribution over various emotion classes. An abstract depiction of the model design is provided in Figure 3.

The input layer (yellow) is followed by four LFBLs. Each LFBL (represented by the next five red blocks) consists of a 2D-CNN, batch normalization, activation, max pooling, and a dropout (20% random drop) layer. The CNN layers within each LFBL are of sizes 64, 64, 128, and 128, and the corresponding max pooling sizes are (2,2), (4,4), (4,4), and (4,4). After the last LFBL, a fully connected layer is used to convert the output of the LFBL into a one-dimensional vector, which is then passed to the LSTM layer (green). The final layer is a dense output layer (blue), whose size corresponds to the number of classes of emotion.

The proposed model achieves an average F1 score of 62%, 59%, and 47% for German, Italian, and Japanese languages, respectively. The exact classes detected by the system may vary depending on the availability of training data in different languages. In the current release, the German variant of the EDS can detect anger, disgust, fear, happiness, neutral, and sadness. In addition to these emotions, the Japanese and Italian variants can also detect surprise.

Implementing the EDS into the e-VITA project is dependent on the upload bandwidth, data minimization, privacy concerns, and prototype environments. Currently, the EDS has been implemented as a component within the Digital Enabler cloud environment, but could alternatively also be implemented on a local edge-computing device with yet-to-be-defined hard-/software specifications. Note that the current implementation of the EDS analyses speech-based audio data exclusively, whereas future versions will integrate processing of video signals to add additional information about the emotional state of the user.

### 3.3. Knowledge Sources

e-VITA has access to a range of knowledge sources, including databases, internet resources, sensor information, emotion detection, common knowledge, and knowledge about the user. The Rasa Open-Source Conversational AI framework, which is used to implement the e-VITA dialogue system, uses various types of knowledge source via its powerful Rasa SDK component (https://rasa.com/docs/rasa-enterprise/) (accessed on 5 January 2023). To access these knowledge sources, the developer can write specific custom actions or use Rasa’s knowledge base actions that support especially two kinds of knowledge source: knowledge graphs and knowledge bases.

Knowledge graphs are used to represent the knowledge that is employed by intelligent AI systems [31]. A knowledge graph (KG) stores data about real-world objects, events, and their relations using nodes and edges, which thus form a network of knowledge for a specific genre or topic. The nodes in the KG represent entities, and the edges represent the properties or relations that link the nodes together. An entity can contain a semantic description of its characteristics, and the links that relate the entities can be based on a different categorization of the world, resulting in networks representing ontologies and type hierarchies besides individual relations between instances of the real world objects.

In the e-VITA project, the KG stores data about the user and the environment that is relevant to active and healthy ageing and that enables the virtual coach to offer personalized information and recommendations. A localized KG stores personal information, e.g., the user’s individual preferences, while a central KG stores information about general aspects of the interaction that are required for various functions and that are to be used across the platform. The central KG is connected to a database that stores procedural details that cannot be harnessed through KG triples. The local KG is on-premise and keeps the personal data safe, while being able to offer assistance and a personalized user experience to an individual user.

The Rasa Conversational AI supports knowledge graph technology with the help of specific knowledge-based actions that supply content for the system responses by querying the domain knowledge base [32]. For instance, part of the knowledge graph for the medical prevention domain is shown in Figure 4. It shows, for example, that recommended medical examinations are age-specific and gender-specific, and also shows information about some of the different examinations related to male and female patients. This kind of knowledge graph was modelled as a Neo4j type labelled property graph [33] (https://neo4j.com/) (accessed on 5 January 2023) and queried using the Cypher query language [34].

However, building knowledge graphs turned out to be more difficult than anticipated, and for quick prototyping we opted for JSON-type knowledge bases because of their simplicity and clarity. An extract from a JSON knowledge base for diseases is shown in Figure 5. For example, if we talk about diseases, the entities will be the specific disease types, while the properties (or attributes) of these entities can include “definition”, “cause”, “symptoms”, “precautions”, “cures”, “risks”, etc., as shown in Figure 5.

Rasa’s knowledge-based actions support question-answer dialogues. For example, the user can ask a question about one of the attributes in the knowledge base, such as “cause” or “prevention” in the example in Figure 5, and the system responds with the text of the corresponding value.

### 3.4. Dialogue System

The Dialogue System is the key component of the virtual coach that enables interactions with the user. Its functions include understanding the user’s input messages (intent recognition), maintaining context (dialogue state tracking), deciding what to do next (dialogue policy), and providing an appropriate response in the given context (response generation) [35]. The context, which is relevant for maintaining coherence in longer-term interactions, includes the previous dialogue context, i.e., what has been talked about earlier, and also the environmental context in terms of sensor information. The dialogue context can range from no context to several turns earlier, and can also take into account previous dialogues that the user has had with the system. The latter option is under development since it requires storing of the dialogues in the system’s memory, which involves not only technical considerations in terms of memory space and retrieval algorithms, but also privacy and trustworthiness aspects concerning what kind of information is saved during the coaching dialogues [3]. The current system only uses the ongoing dialogue context, and experimental research has shown that a history length of three turns provides the best results.

In the e-VITA project, the Dialogue Manager is implemented using the Rasa Open Source Conversational AI framework. Rasa supports a machine learning-based Natural Language Understanding pipeline for the interpretation of the user’s inputs and a combination of rule-based and machine learning-based dialogue policies to determine the system’s actions [36]. It uses Dual Intent and Entity Transformers (DIET) for natural language understanding and Transformer Embedding Dialogues (TED) for system responses.

## 4. The Prototype Virtual Coach

An early prototype version of the virtual coach was delivered at month 15 of the project. This section outlines how the requirements for the prototype were gathered and analysed, followed by a discussion of the use cases addressed in a first feasibility study of the prototype and an example of an interaction with the virtual coach.

### 4.1. Requirements for Gathering and Analysis

Requirements for the virtual coach were gathered in semi-structured interviews with 58 older community-dwelling adults in Germany, Italy, France, and Japan aged 65 and over, with the aim of obtaining information about how older adults organize their daily living and, in particular, with reference to the current work, how they would envisage interactions with a virtual coach [37]. In Germany, 11 older adults, with an average age of 70 years, ranging from 65 to 79, were recruited from a local senior citizens club and acquaintances of project members. In Italy, 5 older adults with a mean age of 70 years were recruited through internal contacts of the National Institute for the Care of the Elderly (INRCA) research unit (https://www.reteneuroscienze.it/en/Istituto/inrca-irccs-istituto-nazionale-di-ricovero-e-cura-per-anziani/) (accessed on 5 January 2023). In France, 12 older adults, average age 74.2 years, were recruited from participants of the workshop “multimedia coffee’ of the Broca Living Lab (https://en.brocalivinglab.org/) (accessed on 5 January 2023). In Japan, 30 adults were recruited from existing contacts who had expressed their desire to be available for future experiments at the university, and from paper advertisements placed in the local city district ward offices. There were 19 female and 11 male participants with an average age of 71.1 (std 2.96). Four lived alone, while 26 lived with family members. Participants had a high interest in new technologies, but generally had little exposure to smart devices and robots. All of the adults in the pre-studies described themselves as regular users of smartphones and personal computers, living independently and not in need of care.

Due to restrictions imposed at the time by the COVID-19 pandemic, semi-structured telephone interviews were used to collect data about what a virtual coach in the form of a social robot should look like, what capabilities it should have, and what topics the participants might wish to talk about with the virtual coach. Generally, a robot that was human-like but not too realistic was preferred and usage scenarios that were discussed included social life, smart home issues, physical support, information provision, nutrition, and reminders.

End-user studies involving actual interactions with real devices were carried out using a process of participatory design in Living Labs in Germany and Japan. In Germany, there were 13 participants, with ages ranging from 65 to 86. In Japan, the Living Lab studies were restricted due to the Corona Emergency Status and all activities took place at the Living Lab of Tohoku University with 4 participants, with ages ranging from 70 to 76. One of the studies conducted in Germany involved interactions with a nutrition chatbot and with the Nao robot, in which a range of scenarios were explored including reminder functions, news, stories, and jokes, and general companionship [38].

The end-user studies in Japan also included interactions with real devices in Living Labs. One of these studies investigated users interacting with the Nao robot and a Rasa dialogue system, which included a general conversation of 10 min with the coach about daily living and a further 10-min conversation about food and general question-answering about the news. The users were instructed to engage in several scenarios, for example:*Tell that you feel sad*—the user expresses sad emotions and the coach aims to provide empathic and consoling companionship;*Tell that you want some exercise*—the user tells the coach that he/she wants some exercise. The coach records the user’s exercise preferences and provides options that can help improve the user’s health and physical condition.

Results from the studies included comments that the users would like to be able to engage in longer conversations and not just receive short answers to their questions. In some cases, the system received user responses that were not included in the original design and that would have to be included in the machine-learning models of the next version of the system. The implementation of information about the user’s daily routines, events in the user’s calendar, user preferences about topics such as music, and reminders was also required.

### 4.2. Interactions with the Prototype Dialogue System

Based on the interviews with users, the following use cases were identified for the prototype system to be developed and evaluated in the initial feasibility study:Daily support;Health activity support;Environmental monitoring support;Question-answering over Wikipedia, news;Social activity support.

The following interaction is based on the Rasa story represented in Figure 6, which is activated following the recognition of the user’s utterance “What preventive examinations should I have performed by medical doctors?” as the intent **ask_examination**. The system consults the knowledge graph shown in Figure 4 and as the system does not know the user’s gender in this use case, it asks the user, to which the user replies “male”. The system follows the **has_right_to** link from the “male” node in the knowledge graph, retrieves the required information, and outputs it to the user. A screenshot of the dialogue produced by this story is shown in Figure 7.

In the initial studies, there were issues with the accuracy and coverage of the Natural Language Understanding (NLU) component. These have been addressed as follows:An intelligent grammar and spell checker based on language models has been added to enable cleansing and pre-processing of the user’s incoming utterances;A better dense featurizer in combination with semantic matching has been developed and added to the NLU pipeline in order to achieve better natural language understanding. For example, in a fallback scenario, the system can detect intents much better than with the previous system.

As a result, the NLU accuracy for interpretation of user queries in English has increased by 5.81% over the vanilla Rasa NLU system and exceeds many other state-of-the-art NLU models.

An important requirement for the e-VITA virtual coach is that it should be multilingual and able to interact with users in English, German, French, Japanese, and Italian. Currently, a translation API is used to manage the translation (https://www.deepl.com/en/translator) (accessed on 5 January 2023). This enables the user to change the language dynamically by issuing a request to the dialogue manager. An example is shown in Figure 8. Here, the coach addresses the user in German and the user requests that they should talk in English. The language is changed and the dialogue continues in English.

### 4.3. User Evaluation

The prototype dialogue system was tested in real conditions in feasibility studies in Europe (Germany, France, and Italy) as well as in Japan, with the aim of assessing its usability and highlighting areas for improvement in the next phase of the project. The feasibility studies reported here examined issues of adherence, accessibility, ease of use, and user experience. Healthy older adults were recruited, who would be prepared to have the e-VITA virtual coaching system installed in their homes. The system included coaching devices, sensors, and a smartphone with applications.

A total of 15 users with a mean age of 71.9 years participated in the studies in Europe, while in Japan there were 5 users, aged 67–91. Users were recruited through various means, including contacts with recreational centres and local associations, as well as existing contacts of test persons who had already participated in the earlier pre-study to gather requirements for the system. The MOCA (Montreal Cognitive Assessment) (https://mocacognition.com/) (accessed on 5 January 2023), the GDS (Geriatric Depression Scale) (https://www.apa.org/pi/about/publications/caregivers/practice-settings/assessment/tools/geriatric-depression) (accessed on 5 January 2023) and the SPPB (Short Physical Performance Battery) (https://www.physio-pedia.com/Short_Physical_Performance_Battery) (accessed on 5 January 2023) were used to determine whether a potential participant was eligible, and a medical declaration of no objection had to be submitted before the start of the study. Additional criteria included: no acute or untreated medical problems such as history of syncopal episodes, epilepsy, and vertigo not controlled pharmacologically; no serious dysfunction of the autonomic system; no severe behavioural syndromes not compensated by drugs; no concurrent neurological diseases; no severe systemic diseases with life expectancy <1 year.

Adherence was recorded automatically based on the number of interactions with the virtual coach. Participants who dropped out were excluded. Other measures included a number of standard questionnaires and semi-structured interviews.

The main aim of the feasibility studies was to obtain feedback from target users in order to redesign the prototype for the larger Proof-of-Concept study in the next stage of the project. The results from the feasibility studies in Europe revealed rather negative opinions towards the technology, however, the participants did not show embarrassment or nervousness when using the technology, even in front of other people. Therefore, it will be essential to take into consideration the participants’ feedback and especially their recommendations in order to improve the virtual coach so that it will add value in their daily life and support them in healthy ageing.

In Germany, Italy, and France, users interacted with the Gatebox and Nao devices and the Rasa software. The experiments in Japan consisted of two parts: first-hand experimental interaction with the Nao robot and the dialogues created using the Rasa software by the Artificial Intelligence Co-Operative Research (AIRC) (https://www.airc.aist.go.jp/en/) (accessed on 5 January 2023). There was also a separate interview with questions related to the user’s expectations and experience of the system using the expectation-experience query method as described in Ref. [39]. Expectations cover the users’ views of the system before the actual interaction, while the experience with the real system was captured by the same questionnaire after the interaction. The questionnaires used a 5-point Likert scale. They were loosely based on the SASSI questionnaire [40] asking about the expectations and experience of the users related to their interaction with a robot, about the robot’s appearance, as well as questions about how the users felt after they had interacted with the robot. Some specific questions related to the e-VITA needs were also included, e.g., safety and robot appearance questions. The results also include comments from the end users.

The experimental setup is shown in Figure 9.

Figure 10 shows the number of interactions that were recorded in the various countries and conditions. In Europe the number of interactions with the Gatebox device far exceeded interactions with the Nao robot (Figure 10A). The number of interactions in Italy exceeded those in Germany and France (Figure 10B). Figure 10C shows the number of interactions with the Nao robot in Japan, while Figure 10D shows the distribution of the interactions across the four countries.

Figure 11 shows user responses to the questionnaires in Japan following interactions with the Rasa dialogue system.

Taking the results of the studies as a whole, the reactions of the users were in general positive, indicating that the contents of the dialogues were applicable to daily living and that interaction was easy. The users found that their actual interactions were better than they had previously expected and that the robot was a good communicator.

Considering expectations, the users felt either more or less comfortable following the interactions, and concerns about talking with a coaching device were alleviated through their experience of interacting with such a device. However, many users found the questions related to user profiling and user preferences uncomfortable. It was also emphasized that cultural background is important in the design of the dialogues, as the content should be tailored to the cultural background of the users. The system should also be able to engage in longer dialogues as well as short question-answer exchanges to allow users to introduce new topics and express themselves more naturally.

Regarding future work, and in particular the development of the Proof of Concept version of the virtual coach, the users in the feasibility studies stressed that the dialogue system needs to be improved in terms of its ability to answer participants’ questions more accurately and deal more effectively with situations where it does not understand or cannot provide an answer. In terms of content, various new features were recommended, such as entertainment (games, jokes, music), daily life functions (reminders), seasonally appropriate information on daily life, more precise nutritional information, general knowledge functions, and dealing with emergency situations (e.g., contacting relatives).

## 5. Conclusions and Future Steps

In this paper, we have described early results from the initial stages of the e-VITA project, a three-year joint European (H2020) and Japanese (MIC) research project in which an innovative coaching system is being developed to address the need for smart living solutions for the care of older adults and the promotion of active and healthy ageing. Following a review of recent related work, the technologies and architecture of the virtual coach were described: multimodal data fusion from sensors; emotion detection; knowledge sources; and the dialogue system that supports spoken and text-based interactions with users. The main focus of the paper is on the dialogue system and how it was employed in an initial prototype version of the virtual coach. Requirements for the virtual coach were gathered in a process involving interviews and participatory design, and the system was tested in end-user studies in the homes of older adults. Although the results from the testing phase were fairly negative, useful suggestions for future improvements were obtained, including new features and improved accuracy.

In the next phase of the project, a randomized controlled Proof of Concept study will be developed in 4 countries with 240 participants (including a control group), with a study duration of 6 months and involving community-dwelling older adult participants. During the re-design phase, the virtual coach will be also tested in living labs in order to obtain iterative feedback from users and ensure that the system will meet the needs and requirements of the target users in terms of techniques and contents. For the upcoming home-based trials, the devices will be prepared by the study centres and installed in the homes of the users. A diary and end-user manuals will be provided, and the system will be explained and demonstrated. A human coach from the community will also visit once a week during the trials.

Future technical work that is currently underway includes developing the knowledge graph modelling and further extending the natural language understanding and dialogue management components. With regard to knowledge modelling, new domains of active and healthy ageing, for example, physical, cognitive, and social, will be added and the contents of each domain will be increased so that the user can access a wider range of queries. Culture-specific aspects will be added as needed, and the databases will be localised to include data relevant to individual users. The emotion detection system is being extended to combine data from speech and facial expressions. A textual sentiment analyser has been developed, which, in combination with the emotion detection system, can enable the dialogue system to engage in dynamic behaviours related to the moods and emotions of individual users. The dialogue manager will also be integrated with sensor data to enable interactions based on data from the sensors. An improved and personalized coaching concept is being developed that takes into account different levels of motivation in an individualized coaching cycle, as well as a recommender system with real events in the local community. Finally, the information search using Wikipedia will be extended to enable more elaborate responses to the user’s information search questions, and deep learning libraries in the form of pre-trained language models are being included in the dialogue manager to enable the system to engage in generic chitchat.

In conclusion, the overall objective of the e-VITA project is to promote active and healthy ageing in older adults in Europe and Japan, to contribute to independent living, and reduce risks of social exclusion. The upcoming final Proof of Concept study will provide extensive feedback on how the technical developments currently underway will enhance user experience and thus contribute to the overall objective of the project, and the real-life needs of the older adult participants in the various countries.

## Figures and Tables

**Figure 1 sensors-23-02748-f001:**
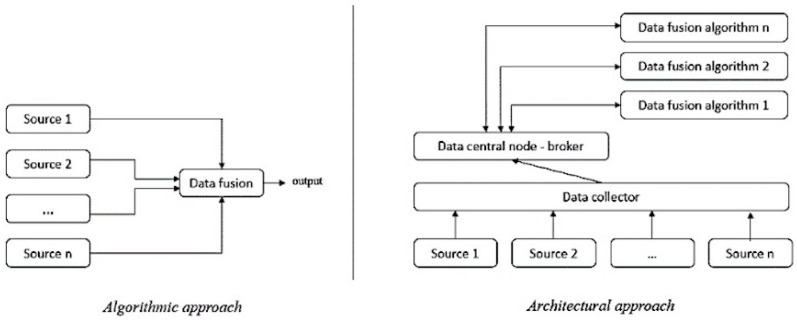
Data fusion from an algorithmic or architectural perspective.

**Figure 2 sensors-23-02748-f002:**
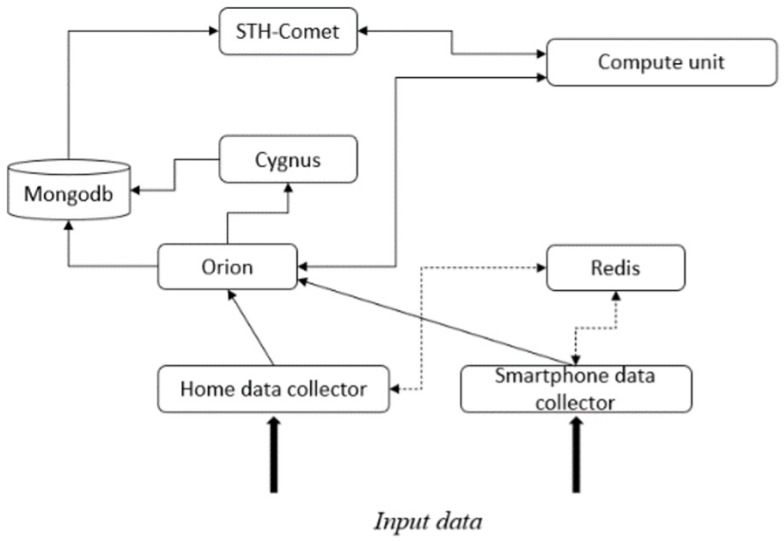
e-VITA data fusion platform architecture.

**Figure 3 sensors-23-02748-f003:**
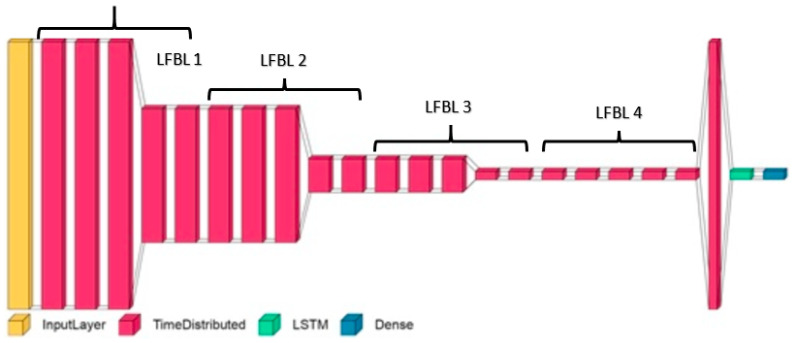
Abstract depiction of the layers in the classifier model used in the EDS.

**Figure 4 sensors-23-02748-f004:**
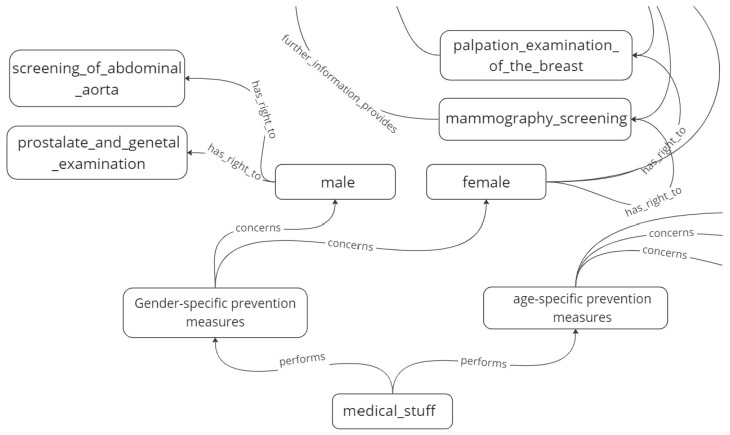
Excerpt from the knowledge graph for the medical prevention domain.

**Figure 5 sensors-23-02748-f005:**
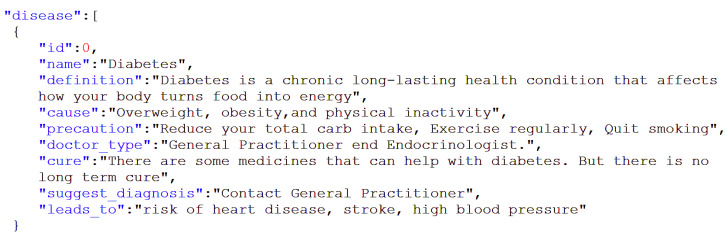
Excerpt from a knowledge base for diseases (which in this case is diabetes).

**Figure 6 sensors-23-02748-f006:**
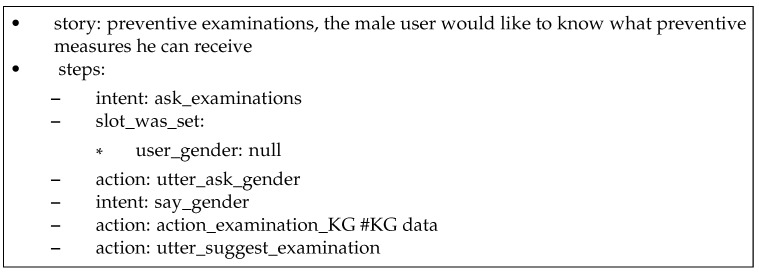
Rasa story: preventive examinations.

**Figure 7 sensors-23-02748-f007:**
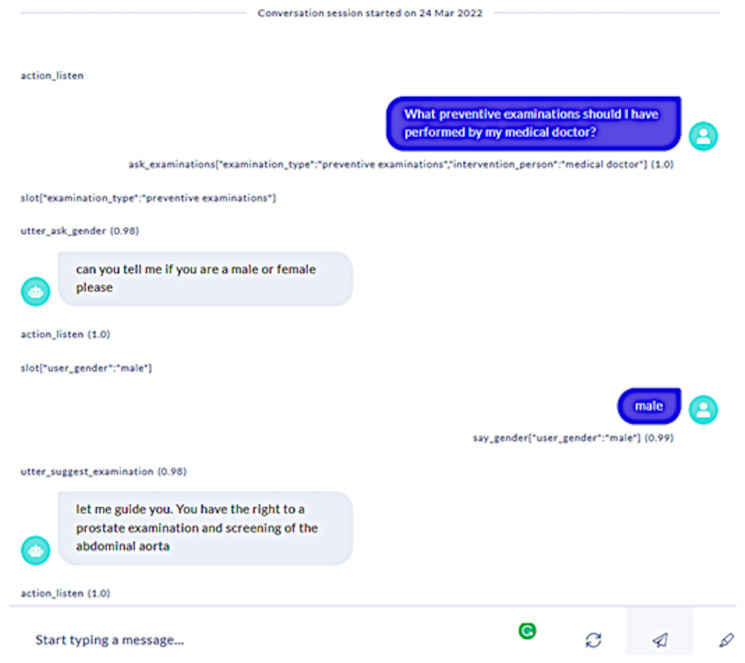
Screenshot of dialogue in the medical prevention domain.

**Figure 8 sensors-23-02748-f008:**
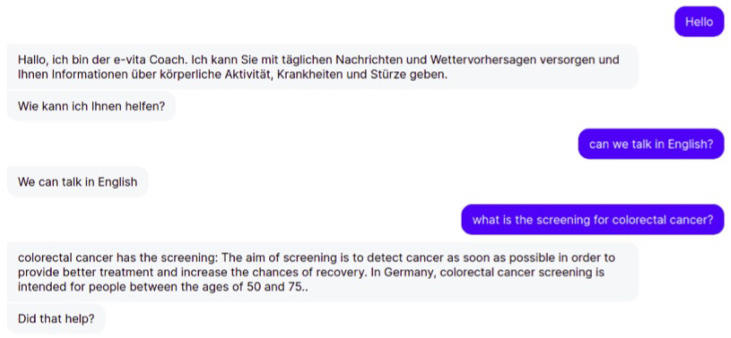
Example of the virtual coach switching language.

**Figure 9 sensors-23-02748-f009:**
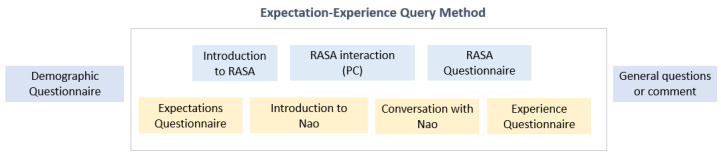
The experimental setup.

**Figure 10 sensors-23-02748-f010:**
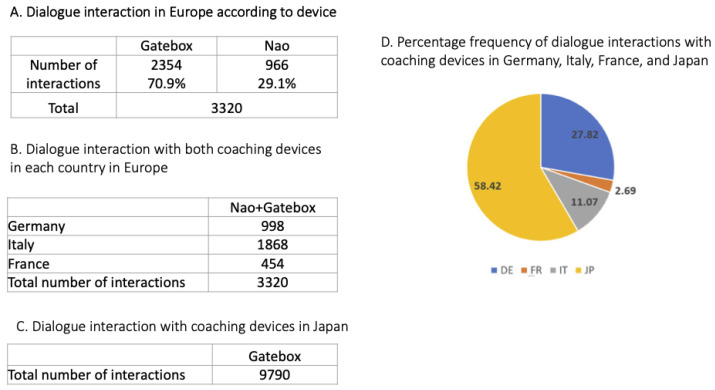
Number of interactions with the devices in the feasibility studies.

**Figure 11 sensors-23-02748-f011:**
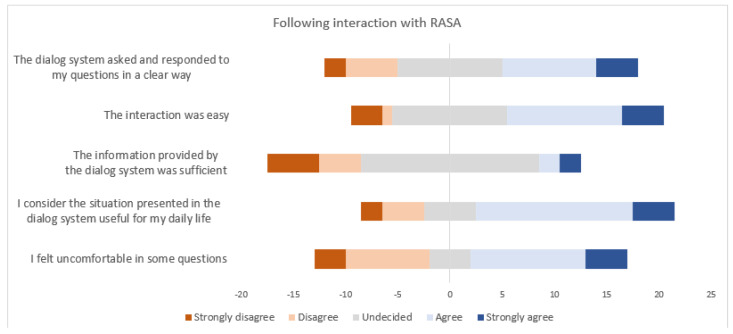
User responses to the questionnaire following interactions with the Rasa dialogue system.

**Table 1 sensors-23-02748-t001:** Classification of sensors.

Type/Name	Event-Driven	Continuous
Worn	Worn	Accelerometer, gyroscope, magnetometer, microphone, etc.
Environment		Thermometer, barometer, etc.
Smart home	PIR, opening detector, etc.	Camera, electric meter, water meter, etc.

## Data Availability

Not applicable.

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
