# Peer review of "Interaction with a Virtual Coach for Active and Healthy Ageing"

_sensors, 2023, doi:10.3390/s23052748_

Round 1
Reviewer 1 Report
Generally, the paper included a complete structure with relatively clear research questions. However, the paper still suffers from insufficientlyrelated works and some language issues, which could be improved before publication.
Language:
The essay has many grammatical errors, mainly the improper use of punctuation and pauses. It is recommended that the article be carefully revised for language problems.
Abstract:
1- The abstract refers to participatory design methods, but the thesis does not describe the process of participatory design in practice.
Related works:
1- Presentation of work in relevant area is too brief.
Prototype Virtual Coach:
4.1
1- What is the number of users, and what are the criteria for recruitment (age, health, etc.)?
2- “In the first phase, users were invited to living labs, while in the second phase, studies were conducted in the homes of users.” The process and content of the two phases must be made clear.
3- The description of the study results is too general and subjective. The results need to be explained with statistical data.
4.3
1- Describe the number of users in the study.
2- What is the meaning of the numbers in the horizontal coordinates of Figure 10?
3- The results of the study in Japan need to be described.
Author Response
Thank you for our paper review and the relevance of comments which will help us to improve the contents of our paper. We agree with the comments concerning the Abstract and related Works sections : we shall improve them accordingly.
About Prototype of virtual Coach, we can indeed add the requested information on Users. Detailed information on process of requirements gathering in living labs and homes is reported in a project deliverable and some of this information will be extracted and inserted into section 4.1. The evaluation studies conducted in Japan are reported in the article [24]. Some additional details will be inserted into section 4.3 from this article.
Concerning statistical data based analysis, as explained in the paper, the work presented here is an intermediate stage of our final prototype to be evaluated in course of this Year 2023, namely from May till August 2023. From that we shall be able to provide deeper and siatistical evaluation on our virtual coach.
In figure 10, indeed the legend does not explain that it is an evaluation score from negative (bad) to positive (best) values for each of the criteria assessed (lines).
Concerning the English text it was internally reviewed and corrected by one of the authors who is a native speaker of English. The final version will be carefully reviewed and corrected by the same author.
Reviewer 2 Report
This international multi-collaborative project is groundbreaking and has the potential to improve millions of lives in the future.
The information presented in this paper is informative and insightful, and also acts as a great reference point for past and future studies related to this project.
In terms of strengthening it further, i would like to see the authors offer some more information around the qualitative and quantitative data collection process that occurred (section 4). For example, please reveal the number of interviews that were conducted, how the interviewees were recruited and conducted (one-on-one/face-to-face) and how the data was analysed? (it also says focus groups in the abstract?)
How many surveys were completed? (section 4.3). how were participants recruited for this part of the project?
It is not sufficient to direct the reader to other studies were that information may have been provided. it also needs to be covered in this article. it is hard to evaluate the contribution to knowledge or strength of the findings without that information.
figure 6 - guessing that should be "reduce total carb intake" (not crab) :)
Author Response
First we would like to thank the Reviewer for our paper review and the relevance of comments which will help us to improve the contents of our paper. We particularly appreciated the high interest in our work as formulated by the Reviewer.
We agree with all the comments and we shall improve the paper accordingly.
About Prototype of virtual Coach, we can indeed add the requested information on Users. Detailed information on process of requirements gathering in living labs and homes is reported in a project deliverable and some of this informatioin will be extracted and inserted into section 4.1. The evaluation studies conducted in Japan are reported in the article [24]. Some additional details will be inserted into section 4.3 from this article. Indeed it is carb intake and not carb intake, thank you.
Reviewer 3 Report
The post is focused on an area that is currently relevant. As the authors state, the contribution is rather descriptive, it lacks a theoretical framework (few resources, some are not available), so it would be appropriate to elaborate more on part 2. To define the goal more scientifically and then elaborate on the other parts of the contribution. In addition to the description of the next steps of the project, it is also necessary to complete the conclusion of the contribution.
Author Response
Thank you for your review of our paper and for the comments on its contents. We shall try to improve the next version of our paper accordingly. Just to bring some elements of answers at this stage, the work presented here is an intermediate step of our Virtual Coach prototype evaluated at the first Field Tests session. The final assessment methodology and related results will be further improved and available after the mid-term of this year 2023. In the meantime we can indeed improve section 2 to make clearer our Virtual Coach (VC) positioning with regard to the state-of-the art. But the design and specification phase of our VC prototype was established as well as by Contents and Technical researchers groups of the e-VITA project with a special concern for scientific and technical rigour with Speech processing and Dialogue management specialists from our partnership.
Concerning the English text it was internally reviewed and corrected by one of the author who is a native speaker of English. The final version will be carefully reviewed and corrected by the same author.
Reviewer 4 Report
The present paper depicts a method of virtual coaching, focused on the analysis of the behavior of elder people, aiming to enable older persons to manage their health and daily activities better.
The paper is interesting and well-presented. However, some improvements can be made.
- In the Related Works section, the topic regarding the analysis of the elderly population using visual systems is completely missing. I think this is a very important topic when discussing the behavior and the health of old people, as the literature is filled with works about it. As a suggestion, I leave here some papers as reference:
L. Romeo, R. Marani, T. D’Orazio and G. Cicirelli, "Video Based Mobility Monitoring of Elderly People Using Deep Learning Models," in IEEE Access, vol. 11, pp. 2804-2819, 2023.
N. K. Mangal and A. K. Tiwari, "A review of the evolution of scientific literature on technology-assisted approaches using RGB-D sensors for musculoskeletal health monitoring," Comput. Biol. Med., vol. 132, pp. 1-15, May 2021.
- I would add a Conclusion section that summarizes all the results obtained.
- In the paper, the smart home is often mentioned, but I would add a schematic image that graphically represents how the setup has been implemented.
- I would add results regarding the EDS. I think this system could be better explained, particularly regarding the deep learning algorithms. Figure 4 can be a bit more precise, as in this form it seems to give not so much information.
- Overall, I would add a Table summarizing all the results, maybe considering a metric to better understand the efficiency of the system.
Author Response
First we would like to thank the Reviewer for our paper review and the relevance of comments which will help us to improve its contents. We also appreciated the high interest in our work. We agree with all of the comments and we shall improve the paper accordingly.
Indeed video-based approaches were considered at the early beginning of the project, namely to get non-verbal emotions e.g. through gestures capture or face expressions, but due to the internal project time constraints and the provision of a first Virtual Coach prototype in last mid-year 2022, we focused more on speech-based emotion capture and capture of actimetric behavioural data with mobile and fixed environmental domotic sensors such as accelerometers, gyroscopic and Presence Infrared sensors (PIR). Indeed we can improve our section 2 by referring to the provided references and precising our current targeted works delimitations which do not preclude to extend to visual data for actimetry/posture monitoring as some of the authors did in past projects (e.g. FP7 EU CompanionAble project) for posture and fall detection.
Round 2
Reviewer 3 Report
Comments were incorporated into the article.